



# Influence of mid-latitude Sea Surface Temperature Fronts on the Atmospheric Water Cycle and Storm Track Activity

Fumiaki Ogawa[1,2] and Thomas Spengler[2]

[1]Graduate School of Science, Hokkaido University, Sapporo, Japan
[2]Geophysical Institute, University of Bergen, and Bjerknes Centre for Climate Research, Bergen, Norway

**Correspondence:** Fumiaki Ogawa (fumiaki.ogawa@sci.hokudai.ac.jp)

**Abstract.** The climatological mean turbulent air-sea heat exchange maximises along midlatitude sea surface temperature (SST) fronts that anchor midlatitude storm tracks. This implies a crucial role of the air-sea latent heat exchange along the SST fronts on the atmospheric water cycle and storm tracks through the intensification of atmospheric cyclones and their associated precipitation. We investigate the sensitivity of the atmospheric water cycle to the SST front through a set of aqua-planet experiments. Varying the latitude of a zonally symmetric midlatitude SST front, the midlatitude atmospheric water cycle responds through distinct changes in surface latent heat fluxes, precipitation, as well as atmospheric moisture fluxes, whereas the tropical latitudes remain largely unchanged. As storm tracks are self-maintained through the diabatic generation of eddy available potential energy, the position of the storm track is diabatically anchored at the SST front. While the position of the SST front determines the position of the eddy moisture convergence and thus the diabatic heating that energises the storm track, the underlying SST determines the general strength of the water cycle and thus the intensity of the storm track. The strong connection identified between the eddy moisture flux and the SST front implies a diabatic pathway of latent heating to anchor the storm track along SST fronts.

## 1 Introduction

The climatological mean vertically integrated global atmospheric water budget obeys the balance

$$C = P - E \tag{1}$$

with evaporation ($E$), precipitation($P$), and convergence ($C$) of the vertically integrated moisture flux ($\boldsymbol{F}$) (e.g., Oki et al., 2004; Trenberth et al., 2007; Schneider et al., 2010; Seager et al., 2010; Trenberth, 2011; Hartmann, 2016; Dey and Döös, 2019). Given that $P$ and $E$ maximise, respectively, along and on the equatorward flank of midlatitude sea surface temperature (SST) fronts associated with the confluent regions of warm and cool oceanic western boundary currents (Zolina and Gulev, 2003; Nakamura et al., 2008; Tilinina et al., 2018; Ogawa and Spengler, 2019), the global atmospheric water cycle is most likely highly sensitive to the position and characteristics of SST fronts. While the response of the midlatitude atmospheric circulation to SST fronts has been well studied (e.g., Nakamura et al., 2004, 2008; Sampe et al., 2010; Ogawa et al., 2012), the response of the water cycle and related implications for the position and strength of the storm track are largely unknown.





Hence, we investigate the sensitivity of the global atmospheric water cycle and storm track to the position and strength of SST
fronts.

With $P < E$ in the subtropical latitudes and $P > E$ both in the equatorial as well as the mid latitudes, $\boldsymbol{F}$ needs to transport moisture from the subtropics to both the inner tropics and midlatitudes (Bryan and Oort, 1984). $\boldsymbol{F}$ can be separated into contributions from the time-mean flow ($\boldsymbol{F}_m$) and eddies ($\boldsymbol{F}_e$) (Hartmann, 2016), where the relative contributions of $\boldsymbol{F}_m$ and $\boldsymbol{F}_e$ to $\boldsymbol{F}$ vary with latitude. While $\boldsymbol{F}$ and $C$ in the Tropics are mainly determined by $\boldsymbol{F}_m$, due to the Eulerian mean
characteristics of the Hadley Cell, $\boldsymbol{F}$ and $C$ are dominated by $\boldsymbol{F}_e$ in the midlatitudes due to the prevalence of atmospheric transient eddies (Peixoto and Oort, 1992; Newman et al., 2012). These transient eddies, associated with the midlatitude storm tracks, play a crucial role for the hydrologic cycle in atmosphere-ocean coupled climate simulations (Dey and Döös, 2019), where most of the transient moisture transport occurs in warm conveyor belts associated with these synoptic eddies (Zhu and Newell, 1998; Neiman et al., 2008; Madonna et al., 2014).

Given that storm tracks in the midlatitudes tend to be anchored to mid-latitude SST fronts (Nakamura et al., 2004, 2008; Sampe et al., 2010; Ogawa et al., 2012; Deremble et al., 2012), the position and strength of SST fronts most likely have significant ramifications for $\boldsymbol{F}_e$ and $C$, contributing to the self-maintenance of storm tracks through the latent heat release associated with $P$, which maximises along the SST fronts due to the collocated storm track activity (Hoskins and Valdes, 1990; Nakamura et al., 2008; Papritz and Spengler, 2015). Furthermore, eddy meridional surface wind yields a maximum in
$E$ on the equator-ward flank of midlatitude SST fronts (Zolina and Gulev, 2003; Tilinina et al., 2018; Ogawa and Spengler, 2019). Despite these strong constraints of the eddy-driven atmospheric water cycle to midlatitudes SST fronts, the sensitivity of the water cycle to the position and strength of SST fronts has not been investigated systematically.

Prescribing zonally symmetric midlatitude SST fronts in idealised aqua-plant simulations highlight a strong sensitivity of the zonal mean atmospheric circulation as well as the latitude of the storm track to the position of SST fronts (Ogawa et al.,
2012, 2016). As both low-level atmospheric baroclinicity associated with the SST front (Nakamura et al., 2008; Sampe et al., 2010; Hotta and Nakamura, 2011) as well as latent heat release in midlatitude storms (Hoskins and Valdes, 1990; Deremble et al., 2012; Papritz and Spengler, 2015) have been argued to maintain the location of the storm track, it is important to further clarify the constraints of SST fronts on the atmospheric water cycle and their relevance to the position and intensity of storm tracks. We quantify this sensitivity of the mean meridional atmospheric water cycle and storm track intensity to the existence
and latitude of midlatitude SST fronts using a set of aqua-planet atmospheric general circulation model (AGCM) experiments.

## 2   Aqua-planet experiments

We conduct a set of aqua-planet AGCM experiments with prescribed zonally symmetric SSTs featuring a midlatitude SST front. We use the AGCM of the Earth Simulator (Ohfuchi et al., 2004; Enomoto et al., 2008; Kuwano-Yoshida et al., 2010, AFES) with a triangular truncation of T79, corresponding to $\approx 1.5°$ grid spacing in longitude and latitude. Though the resolution is
insufficient to resolve mesoscale features of the SST, the resolution is sufficient to realistically represent both the climatological-mean state and annular mode variability (Nakamura et al., 2008; Sampe et al., 2010, 2013; Ogawa et al., 2012, 2015, 2016). The





model has 56 $\sigma$-levels in the vertical that reach up to 0.09 hPa, which is well above the stratopause. The model was integrated for 120 months using the perpetual austral winter solstice condition, where we disregard the first six months as spin-up.

The default SST profile is based on the observed SST averaged over the South Indian Ocean (60°-80°E) using the OISST
data (Reynolds et al., 2007). The observed climatological mean South Indian Ocean SST for austral winter (June-July-August) was assigned to the Southern Hemisphere and the corresponding austral summertime profile (December-January-February) was assigned to the Northern Hemisphere (black solid line in Fig. 1). Henceforth, we will refer to the Southern and Northern Hemisphere as the winter and summer hemisphere, respectively. The SST is identical to the one used in Ogawa et al. (2016).

For the sensitivity experiments, the magnitude of the SST gradient was kept identical while the latitude of the SST fronts
in both hemispheres was shifted to either 35° or 55° (red and blue lines in Fig. 1b). We refer to the experiments as F35, F45, and F55, respectively. In addition, we performed an experiment using a non-front reference SST profile (REF), where the midlatitude SST front is smoothed while maintaining the same SST gradient poleward of the SST front (gray line in Fig. 1). More details about the model setup can be found in Ogawa et al. (2016). The REF experiment is identical to the NF experiment in Ogawa et al. (2016), though is referred to as REF, because of the presence of a weak local maximum of the subtropical SST
gradient around 33° in both hemispheres.

For all experiments, the SST equator-ward of 25° is unchanged (Fig. 1a) to avoid a direct impact on the Hadley cell (Höjgård-Olsen et al., 2020), which would yield a remote influence on the midlatitude jets (Watt-meyer and Frierson, 2019). The experiments in this study are not meant to reproduce the observed atmospheric circulation, but to clarify the impact of the midlatitude SST front on the atmospheric circulation and the atmospheric water cycle.

In addition, we performed experiments where we uniformly increased or decreased the SST by 5 K for the F45 and REF setup. We refer to these as F45+5 and F45-5 for simulations with the SST front at 45° and a 5 K higher and lower SST, respectively. Similarly, we refer to REF+5 and REF-5 for simulations with the reference state without a SST front and a 5 K higher or lower SST, respectively. The SST gradients for these experiments are identical to F45 and REF (see Fig. 1b), while the SST is uniformly higher or lower, respectively (not shown).

We use 6-hourly instantaneous fields of surface pressure ($p_s$), specific humidity ($q$), evaporation ($E$), convective precipitation ($P_c$), large-scale precipitation ($P_l$), zonal ($u$) and meridional wind ($v$), vertical pressure ($\omega$) and sigma velocity ($\dot{\sigma}$), as well as 6-hourly integrated values of changes in temperature ($\dot{Q}$) and specific humidity due to parameterisations ($\dot{q} = \dot{q}_c + \dot{q}_m + \dot{q}_b$), which is split up in tendencies from convection parameterisation ($\dot{q}_c$), microphysics parameterisation ($\dot{q}_m$), and boundary layer parameterisation ($\dot{q}_b$). We use data at all available $\sigma$ levels.

## 3   Diagnosing the atmospheric water cycle

We use the tendency equation for specific humidity (2) and continuity in sigma coordinates (3)

$$\frac{\partial q}{\partial t} + \frac{u}{a\cos\phi}\frac{\partial q}{\partial \lambda} + \frac{v}{a}\frac{\partial q}{\partial \phi} + \dot{\sigma}\frac{\partial q}{\partial \sigma} = \dot{q} \tag{2}$$

$$\frac{\partial p_s}{\partial t} + \frac{1}{a\cos\phi}\frac{\partial(p_s u)}{\partial \lambda} + \frac{1}{a\cos\phi}\frac{\partial(p_s v\cos\phi)}{\partial \phi} + \frac{\partial(p_s \dot{\sigma})}{\partial \sigma} = 0, \tag{3}$$





to derive the flux form for specific humidity

$$\frac{\partial (p_s q)}{\partial t} + \frac{1}{a \cos\phi} \frac{\partial (p_s u q)}{\partial \lambda} + \frac{1}{a \cos\phi} \frac{\partial (p_s v q \cos\phi)}{\partial \phi} + \frac{\partial (p_s \dot\sigma q)}{\partial \sigma} = p_s \dot{q}, \tag{4}$$

where the wind vector in sigma coordinates is $\boldsymbol{v} = (u, v, \dot\sigma)$, $q$ is specific humidity, $p_s$ is surface pressure, $a$ is the radius of Earth, and $\phi$ and $\lambda$ are the latitude and longitude, respectively.

Due to the zonal symmetry of the experiments, we use (4) to derive the atmospheric water cycle for the zonally and time averaged climatological mean state

$$\nabla \cdot \boldsymbol{F} = \frac{1}{a\,g\cos\phi} \frac{\partial \left(\overline{[p_s v q]}\cos\phi\right)}{\partial \phi} + \frac{1}{g} \frac{\partial \overline{[p_s \dot\sigma q]}}{\partial \sigma} = \frac{\overline{[p_s \dot{q}]}}{g}, \tag{5}$$

where $\overline{\frac{\partial (p_s q)}{\partial t}} = 0$, $g$ is the gravitational constant, $\boldsymbol{F} = \frac{1}{g}\left(0, \overline{[p_s v q]}, \overline{[p_s \dot\sigma q]}\right)$ is the moisture flux, and the squared brackets $[f]$ and the bar $\overline{f}$ denote the zonal average and time mean, respectively. Given the dominant contribution of the eddy moisture fluxes in the extra-tropics (e.g., Peixoto and Oort, 1992; Newman et al., 2012), we separate the variables ($f = \overline{f} + f'$) into a time mean ($\overline{f}$) and eddy ($f'$) component yielding

$$\nabla \cdot \boldsymbol{F} \approx \nabla \cdot (\boldsymbol{F}_m + \boldsymbol{F}_e), \tag{6}$$

with $\boldsymbol{F}_m = \frac{1}{g}\left(0, [\overline{p_s}\,\overline{v}\,\overline{q}]\,[\overline{p_s}\,\overline{\dot\sigma}\,\overline{q}]\right)$ and $\boldsymbol{F}_e = \frac{1}{g}\left(0, \left[\overline{p_s}\,\overline{v'q'}\right], \left[\overline{p_s}\,\overline{\dot\sigma'q'}\right]\right)$ being the mean and eddy components, where we excluded terms proportional to $\overline{q}\overline{p_s' v'}$, $\overline{v}\overline{p_s' q'}$, $\overline{q}\overline{p_s' \dot\sigma'}$, $\overline{\dot\sigma}\overline{p_s' q'}$, $\overline{\dot\sigma'p_s' q'}$, and $\overline{v'p_s' q'}$, as they turned out to be negligible (not shown).

Integrating (5) vertically, letting the fluxes at the surface to be precipitation ($P$) and evaporation ($E$), yields the atmospheric water cycle

$$C = P - E = P_l + P_c - E, \tag{7}$$

where $P_l$ and $P_c$ are large-scale and convective precipitation, respectively, and

$$C \approx C_m + C_e = -\int_0^1 \nabla \cdot (\boldsymbol{F}_m + \boldsymbol{F}_e)\,d\sigma = -\frac{1}{a\,g\cos\phi} \frac{\partial}{\partial \phi} \int_0^1 \left([\overline{p_s}\,\overline{v}\,\overline{q}]\cos\phi + [\overline{p_s}\,\overline{v'q'}]\cos\phi\right) d\sigma, \tag{8}$$

is the climatological mean vertically integrated atmospheric moisture flux convergence, where we used $\dot\sigma = 0$ at $\sigma = 0/1$. The first and second term in (8) indicate the moisture flux convergence by the mean ($C_m$) and eddy ($C_e$) flux, respectively.

## 4  Energy framework for the storm track

The climatological mean energy cycle can be diagnosed using the Lorenz framework (Lorenz, 1955; Oort and Peixoto, 1974; Peixoto and Oort, 1974), where the contribution of latent heating can be diagnosed as the diabatic eddy energy generation. Following Peixoto and Oort (1974), we use the climatological mean and zonally averaged atmospheric energy as well as its





conversion terms defined on the two-dimensional latitude-pressure domain

$$\text{APE}_Z = \frac{c_p}{2}\gamma\left[\overline{T}\right]^{\#2} \tag{9}$$

$$\text{APE}_E = \frac{c_p}{2}\gamma\left[\overline{T'^2} + \overline{T}^{*2}\right] \tag{10}$$

$$\text{KIN}_Z = \frac{1}{2}([\overline{u}]^2 + [\overline{v}]^2) \tag{11}$$

$$\text{KIN}_E = \frac{1}{2}\left[\overline{u'^2} + \overline{v'^2}\right] + \frac{1}{2}\left[\overline{u}^{*2} + \overline{v}^{*2}\right] \tag{12}$$

$$\text{CE} = -\frac{R}{p}\left[\overline{\omega'T'} + \overline{\omega}^*\overline{T}^*\right] \tag{13}$$

$$= \frac{R}{p}[\overline{\omega}]^{\#}\left[\overline{T}\right]^{\#} \tag{14}$$

$$\text{CA} = -c_p\gamma\left[\overline{v'T'} + \overline{v}^*\overline{T}^*\right]\frac{\partial[\overline{T}]}{a\partial\phi}$$
$$-c_p p^{-\kappa}\left[\overline{\omega'T'} + \overline{\omega}^*\overline{T}^*\right]\frac{\partial\left(\gamma p^{\kappa}\left[\overline{T}\right]^{\#}\right)}{\partial p} \tag{15}$$

$$\text{CK} = \left[\overline{v'u'} + \overline{v}^*\overline{u}^*\right]\cos\phi\frac{\partial([\overline{u}]/\cos\phi)}{a\partial\phi}$$
$$+ \left[\overline{v'^2} + \overline{v}^{*2}\right]\frac{\partial[\overline{v}]}{a\partial\phi}$$
$$+ \left[\overline{\omega'u'} + \overline{\omega}^*\overline{u}^*\right]\frac{\partial([\overline{u}])}{\partial p}$$
$$+ \left[\overline{\omega'v'} + \overline{\omega}^*\overline{v}^*\right]\frac{\partial([\overline{v}])}{\partial p}$$
$$- [\overline{v}]\left[\overline{u'^2} + \overline{u}^{*2}\right]\frac{\tan\phi}{a} \tag{16}$$

$$\text{GE} = \gamma\left[\overline{Q'T'} + \overline{Q}^*\overline{T}^*\right] \tag{17}$$

$$\text{GZ} = \gamma\left[\overline{Q}\right]^{\#}\left[\overline{T}\right]^{\#}, \tag{18}$$

with $\gamma = -\left(\frac{R\theta}{T c_p p}\right)\left(\frac{\partial\Theta}{\partial p}\right)^{-1}$, $\theta = T\left(1000/p\right)^{\kappa}$, $\kappa = \frac{R}{c_p}$, where $R$ is the gas constant, $c_p$ is the specific heat at constant pressure, $\Theta$ is the global horizontal average of $\theta$, superscripts $^*$ and $^{\#}$ denote anomalies from zonal and meridional averages, respectively. Eddy and mean-flow components in APE and KIN are denoted by the subscripts $_E$ and $_Z$, respectively.

The three-dimensional spatial mean energy budget can be expressed by

$$d\langle\text{APE}_Z\rangle/dt = -\langle\text{CA}\rangle + \langle\text{CZ}\rangle + \langle\text{GZ}\rangle - \langle\text{DAPE}_Z\rangle \tag{19}$$

$$d\langle\text{APE}_E\rangle/dt = \langle\text{CA}\rangle - \langle\text{CE}\rangle + \langle\text{GE}\rangle - \langle\text{DAPE}_E\rangle \tag{20}$$

$$d\langle\text{KIN}_E\rangle/dt = \langle\text{CE}\rangle - \langle\text{CK}\rangle - \langle\text{DKIN}_E\rangle \tag{21}$$

$$d\langle\text{KIN}_Z\rangle/dt = \langle\text{CK}\rangle - \langle\text{CZ}\rangle - \langle\text{DKIN}_Z\rangle, \tag{22}$$

where $\langle f\rangle$ indicates the average over the meridional plane. The left-hand side must vanish climatologically, so the dissipation term, which appears as the last term on the right-hand side in (19)-(22), was calculated as the residual to close the energy budget. We interpolate the data from $\sigma$ to pressure levels and calculate vertically mass weighted averages from 850hPa to





200hPa. We obtain a reasonable closure of the energy cycle, where the total of CA and GE is very close to CE, which supplies the eddy kinetic energy ($\text{KIN}_E$).

We restrict our discussion to the energy conversions relevant to the storm track activity: conversion of available potential energy from the zonal mean state to eddies (CA), baroclinic eddy energy generation (GE), baroclinic eddy energy conversion (CE), and the eddy kinetic energy ($\text{KIN}_E$).

## 5 Vertically integrated water cycle

The climatological mean vertically integrated atmospheric moisture cycle shows the gain and loss of atmospheric water content as a function of latitude according to (7) and (8), where the sum of the five different terms is approximately zero at every latitude, confirming climatological balance (Fig. 2). At lower latitudes, moisture is provided by $E$ from the warm tropical ocean, which is subsequently converged into the inter-tropical convergence zones (ITCZs) in the summer and winter hemisphere by $C_m$, where $P_c$ removes the moisture from the atmosphere. Two latitudinal peaks of $P_c$ and the latitude of mean upward motion in the Tropical latitudes (Fig. 3c-d,S1c-d) indicate the existence of a double-ITCZ, which is presumably caused by the prescribed tropical SST (Watt-meyer and Frierson, 2019). The stronger and more equator-ward peak in $P_c$ in the winter hemisphere compared to the summer hemisphere reflects the prescribed season. The similarity of the moisture cycle in the lower latitudes across all experiments (Fig. 2a-d) suggests that the position and existence of the midlatitude SST front have little influence on the Hadley circulation (Fig. 3c,d, S1c,d).

In the midlatitudes, however, the atmospheric moisture budget is sensitive to the latitudinal position as well as the existence of an SST front. Both $C_e$ and $P_l$ maximise on the poleward flank of the SST front (black and blue dashed lines in Fig. 2a-c), which is consistent with the positioning of the storm track (Ogawa et al., 2012, 2016). $E$ features a stark contrast across the SST front (red lines in Fig. 2a-c), with a notable increase from the poleward toward the equator-ward side of the SST front. This increase is mostly due to the surface saturation-specific humidity as a function of SST through the Clausius Clapeyron relation and not the mean surface wind speed (not shown). Similarly, $P_c$ (blue solid lines in Fig. 2a-c) also features higher values on the equator-ward side of the SST front. $C_m$ is generally less than $C_e$ near the SST front (solid and dashed black lines in 2a-d), highlighting the dominance of eddy contributions around the SST front. While the amount of air-sea moisture exchange is slightly less in the summer hemisphere, the SST front still strongly influences the atmospheric water budget in both hemispheres.

In REF (Fig. 2d), the absence of the SST front results in smoothed profiles in the extra-tropics without any distinctive peaks or steep transitions as found in the experiments with an SST front (Fig. 2a-c). Given the absence of a peak in $C_e$, the moisture convergence by midlatitude eddies is significantly reduced, which is consistent with a suppression of storm track activity in the absence of an SST front (Nakamura et al., 2008; Sampe et al., 2010, 2013). Comparing REF to the front experiments, it is also evident that the evaporation is strongly influenced by absolute SST, which is due to the Clausius Clapeyron dependence of the saturation mixing ratio at SST.





## 6 Meridional overturning of atmospheric moisture

The meridional overturning of atmospheric moisture depicted by $\boldsymbol{F}$ reveals a clear separation of the tropical and extratropical
cells for F45 (Fig. 3a,c). The moisture converges into the ITCZs in the tropical latitudes, where it is transported upward by
the mean circulation. The subsiding branch in the subtropical latitudes splits in an equator-ward and a poleward component,
with significant divergence in the lower troposphere. The direction of $\boldsymbol{F}$ in the midlatitudes features a clear poleward ascent,
which is associated with the isentropic up-glide along the isentropes in the warm conveyor belts in extratropical cyclones
(McTaggart-Cowan et al., 2017).

In the tropical latitudes, the meridional overturning of atmospheric moisture is dominated by $\boldsymbol{F}_m$ (Fig. 3c), which is consistent with $C_m$ being the main contributor to the divergence in the subtropics and convergence in the ITCZs (red line Fig.
2b). Thus, the tropical circulation largely resembles the Hadley cell with the inner-tropical $\boldsymbol{F}$ being almost entirely explained
by $\boldsymbol{F}_m$. In the vicinity of the extratropical SST front, the amplification of $\boldsymbol{F}_m$ in the poleward direction near the surface is
associated with the thermally indirect Ferrel cell (Fig. 3c).

The extratropics are dominated by $\boldsymbol{F}_e$, which is ascending poleward along isentropic surfaces (Fig. 3a), consistent with the
moisture fluxes along warm conveyor belts associated with midlatitude cyclones (Madonna et al., 2014; Papritz and Spengler,
2015; McTaggart-Cowan et al., 2017). The poleward ascending flux implies a divergence in the lower troposphere equator-
ward of the SST front and a convergence in the mid to upper troposphere in the midlatitudes, which is more or less centred on
the SST front. Consistently, $C_e$ peaks on the poleward side of the SST front, yielding a peak in $P_l$ associated with the warm
conveyor belts (Fig. 2a).

Varying the position of the SST front yields qualitatively similar results (Fig. S1). In the tropical latitudes, $\boldsymbol{F}_m$ and its
associated divergence/convergence are almost identical across all experiments, whereas $\boldsymbol{F}_e$ and its associated divergence shift
with the latitude of the SST front for F35, F45, and F55 in the lower troposphere (Fig. 3, S1). For REF, the absence of the
SST front results in a wider latitudinal spread of the lower tropospheric divergence of $\boldsymbol{F}_e$, stretching from the subtropics to
the higher latitudes (Fig. 3b). The mid and upper tropospheric convergence of $\boldsymbol{F}_e$ is generally less affected by the position and
existence of the SST front and mainly widens with a higher latitude SST front, or if no SST front is prescribed. The latter is
also evident in a more widely spread $P_l$ for F45 and F55 as well as for REF (Fig. 2).

## 7 Moisture flux convergence and parameterised processes

To achieve climatological balance in (5), the convergence of $\boldsymbol{F}$ (Fig. 3a) is counteracted by $\dot{q}$, which consists of the micro-
physics parameterisation ($\dot{q}_m$), the convection parameterisation ($\dot{q}_c$), and the boundary layer parameterisation ($\dot{q}_b$) (Fig. 4a,c,e).
The distribution of $\dot{q}_m$ and $\dot{q}_c$ are strikingly different, where the vertical integrals of $\dot{q}_m$ and $\dot{q}_c$ (Fig. 4e,g) correspond to $P_l$ and
$P_c$, respectively (Fig. 2b).

$\dot{q}_m$ is mainly located in the midlatitude mid-troposphere, which solely contributes to the removal of moisture from the
atmosphere (Fig. 4a) and is balanced by the convergence of $\boldsymbol{F}_e$ (Fig. 3a). The moisture removal occurring along the isentropic
slope, which is in thermal wind balance with the jets (Fig. 4a), corresponds to the precipitation formed along the up-gliding of





moist air-parcels in cyclones (Papritz and Spengler, 2015). $\dot{q}_c$, on the other hand, is balanced by the convergence of $\boldsymbol{F}_m$ (Fig. 3c) and features losses of atmospheric moisture in the planetary boundary layer at all latitudes as well as within the vertical towers of the ITCZ (Fig. 4c). $\dot{q}_c$ in the subtropical lower troposphere adds moisture to the atmosphere (Fig. 4c) and is largely compensated by the divergence of $\boldsymbol{F}_m$ (Fig. 3c).

The secondary peak of $P_c$ in the midlatitudes (Fig. 2b) is associated with an increasing vertical extent of negative $\dot{q}_c$ from the tropical latitudes towards the SST front, where $\dot{q}_c$ drops dramatically (Fig. 4c). The gains of $\dot{q}_c$ in the subtropical mid-troposphere are associated with a vertical rearrangement of moisture by the convection scheme (Fig. 4c) and are compensated by the divergence of $\boldsymbol{F}_e$ and $\boldsymbol{F}_m$ (Fig. 3a and 3c).

        The contribution of $\dot{q}_b$ represents the turbulent uptake of moisture from the ocean and its redistribution in the atmospheric
boundary layer (Fig. 4i). The vertical integral of $\dot{q}_b$ corresponds to $E$ (red line in Fig. 2b) and accordingly features a stark contrast across the SST front. The moisture supplied by $\dot{q}_b$ is largely compensated by $\dot{q}_c$ (Fig. 4c,e). The areas of high climatological mean moisture amount in the subtropical planetary boundary layer appear to be largely determined by regions where $\dot{q}_b$ is large (Fig. 4e). For REF there is no significant drop in $\dot{q}_b$ and $\dot{q}_c$ at the SST front, instead they gradually decrease towards higher latitudes (Fig. 4b,d).

Similar to $\boldsymbol{F}$, $\dot{q}$ in the lower latitudes is not sensitive to the position or existence of the SST front (Fig. S2). However, the stark contrast in vertical extent of $\dot{q}_b$ and $\dot{q}_c$ remains anchored to the SST front when the latter is shifted latitudinally (Fig. S4a-b). Similarly, $\dot{q}_m$ is shifted with the SST front (Fig. S4c-f), which is consistent with the sensitivity of the vertically integrated moisture budget (Fig. 2).

## 8     Diabatic enhancement of storm track activity

The importance of the SST front for both the latitude of the storm track (Ogawa et al., 2012) and the eddy moisture flux convergence shown in this study implies a diabatic enhancement of storm track activity through energising individual storms via surface latent heat exchange (Haualand and Spengler, 2020; Bui and Spengler, 2021). We diagnose the Lorenz energy cycle (Lorenz, 1955; Oort and Peixoto, 1974), where the contribution of the latent heat can be diagnosed as the diabatic eddy energy generation. Considering pressure levels from 850hPa to 200hPa, we obtain a reasonable closure of the energy cycle (Fig. 5),
where the total of CA and GE is very close to CE, the conversion to eddy kinetic energy ($\text{KIN}_E$).

        While the direct impact of diabatic heating by GE ($0.44 \times 10^5 \, \text{J m}^{-2} \, \text{day}^{-1}$) is only $\approx 15\%$ of CA, the diabatic heating also alters the temperature anomaly and thus contributes implicitly to CA. Thus, the ratio between the hemispherically averaged GE and CA cannot directly suggest a relative impact of moisture for the storm track activity. Nevertheless, given the systematically organised eddy moisture flux convergence (Fig. 3) as well as the corresponding diabatic heating (Figs. 4), it is worthwhile to
235 discuss the meridional distributions of the vertically averaged energy budget to obtain an indication for the local impact of moisture in the energy budget.

        Consistent with Ogawa et al. (2012), the peak latitude of CA is sensitive to the position of the SST front (Fig. 6a). While CE, GE, and $\text{KIN}_E$ increase for an equator-ward shift of the SST front (Fig. 6b-d), CA remains largely unchanged (Fig. 6a). The





sensitivity of GE is consistent with higher SST yielding enhanced diabatic heating around the SST front in the mid-troposphere

(red lines in Fig. 2). The reduction of GE (Fig. 6c) along the equatorward flank of the SST front is related to surface heat fluxes dampening near-surface atmospheric temperature anomalies. The enhancement of GE on the poleward flank of the SST front, on the other hand, is due to mid-tropospheric diabatic heating associated with the eddy moisture flux convergence (Fig. 3). In fact, the double peak structure in GE (Fig. 6c) is hinted in both CA and $KIN_E$ (Fig. 6a,d), suggesting a considerable impact of diabatic processes on both $KIN_E$ and CA.

The lower-latitude peaks in CA, GE, CE, and $KIN_E$ are always located at a similar latitude near $33°$ for F45, F55, and REF, where GE in REF is as strong as F45 and F55 (Fig. 6). This is consistent with the eddy moisture flux convergence peaking at a similar latitude (Fig. 2b-d). These similarities of the latitudinal peaks in both storm track activity and eddy moisture flux convergence are consistent with the existence of a secondary maximum in the SST gradient around $33°$ for F45, F55, and REF (Fig. 1b). For F35, this weak subtropical SST gradient is joint with the main SST front at $35°$ and the storm track activity and

water cycle maximise at the poleward flank of the main SST front.

To disentangle the impact between SST and SST gradient, we repeat F45 and REF with a uniform increase or decrease of SST by 5K, referred to as F45+5, F45-5, REF+5, and REF-5, respectively. The vertically integrated water cycle amplifies (weakens) for higher (lower) SST, without a systematical shift in latitudinal position (Fig. 7). The peak of the moisture flux convergence on the poleward flank of the SST front in F45 amplifies (reduces) by ≈50% for F45+5 (F45-5). This is consistent

with a change in the mean moisture flux convergence in conjunction with the strength of the Hadley circulation, balanced by convective condensation, as well as changes in eddy moisture flux convergence together with the steepness of isentropic surfaces, balanced by microphysics condensation (Figs. S3-6).

The uniform changes of SST also modify the storm track activity, yielding enhanced (reduced) storm track activity for higher (lower) SST ($KIN_E$; Fig. 8d). Despite small changes in CA (Fig. 8a), GE and CE show a clear increase (decrease) for higher

(lower) SST ($KIN_E$; Fig. 8b,c), consistent with changes in $KIN_E$ (Fig. 8d). The sensitivity of storm track activity to uniform changes in SST (Fig. 8) can hence also explain the sensitivity of the storm track activity to the position of the SST front (Fig. 6), as the mean SST is higher (lower) for F35 (F55).

## 9 Conclusions

Using a set of idealised aqua-planet experiments, we investigated the sensitivity of the atmospheric water cycle and storm track

activity to shifting or removing the midlatitude SST front. When shifting the latitude of the SST front, the tropical circulation and atmospheric water cycle are almost unaffected, with barely any changes in the time-mean circulation and Hadley cell. In the midlatitudes, however, evaporation, convective and large-scale precipitation, as well as the eddy moisture flux and its convergence are shifting latitudinally in concert with the SST front.

The atmospheric water budget in the tropical latitudes is dominated by evaporation from the warm ocean surface, providing

moisture to the boundary layer that is then transported upward by the convection parameterisation in the subtropics. In the inner tropics, the time-mean circulation associated with the Hadley cell provides upward moisture transport and convergence, where



the moisture is released as convective precipitation along the ITCZ. In the midlatitudes, on the other hand, upward surface latent heat fluxes and convective precipitation maximise along the equator-ward flank of the SST front whereas large-scale precipitation and atmospheric eddy moisture flux convergence maximise on the poleward flank of the SST front. The absence of these features in a simulation without an SST front confirms the importance of the SST front for the atmospheric water cycle in the midlatitudes.

As stormtracks are self-maintenaned through the diabatic generation of eddy available potential energy (GE) associated with the eddy moisture flux convergence ($C_e$) and microphysics condensation ($\dot{q}_m$), the position of the storm track is diabatically anchored at the SST front. This is consistent with eddy kinetic energy ($\text{KIN}_E$) and microphysics condensation ($\dot{q}_m$) being weaker and less confined for the experiment without an SST front, indicating that the SST front latitudinally constrains and thereby enhances the storm track activity.

Given that surface latent heat fluxes are largely determined by the surface saturation mixing ratio at SST through the Clausius Clapeyron relation, the distribution of evaporation and convective precipitation are strongly tied to the SST. Hence, moving the SST front to higher (lower) latitudes at relatively lower (higher) SST reduces (increases) the strength of the water cycle and thereby the intensity of the storm track. This sensitivity is confirmed through experiments with globally increased (reduced) SST, with the water cycle and storm track intensity changing by 50% for SST changes of $\pm 5$K.

Our results demonstrate the importance of both SST and SST fronts to understand the midlatitude atmospheric water cycle and storm track activity. While the position of the SST front determines the position of the eddy moisture convergence and thus the diabatic heating that energises the storm track, the underlying SST determines the general strength of the water cycle and thus the intensity of the storm track. Our study thus highlights the importance of correctly representing SST and the position SST fronts to ensure an adequate representation of both the atmospheric water cycle and the storm track.

*Code and data availability.* The data used in this study is publicly available from the NIRD research data archive; https://doi.org/10.11582/2020.00065. Figures shown in this study are plotted using the NCAR Command Language (NCL, https://www.ncl.ucar.edu/). Codes can be obtained from the corresponding author.

*Author contributions.* FO and TS are the co-initiators of this study; FO conducted the analysis and was the first author of the main text; TS guided the assembly of the study and helped revise the paper.

*Competing interests.* The contact author has declared that none of the authors have any competing interests.

*Acknowledgements.* FO was supported by MEXT, a Grant-in-Aid for Scientific Research in Innovative Area 6102 ("Hotspot2" project). TS was supported by the project BALMCAST (Research Council of Norway project number 324081). This study was supported by UNPACC



(Research Council of Norway project number 262220). We also thank Ophélie Nourrisson and Florian Marris for some initial work on this topic. Computing resources were provided by Japan Agency for Marine Science and Technology (JAMSTEC).



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

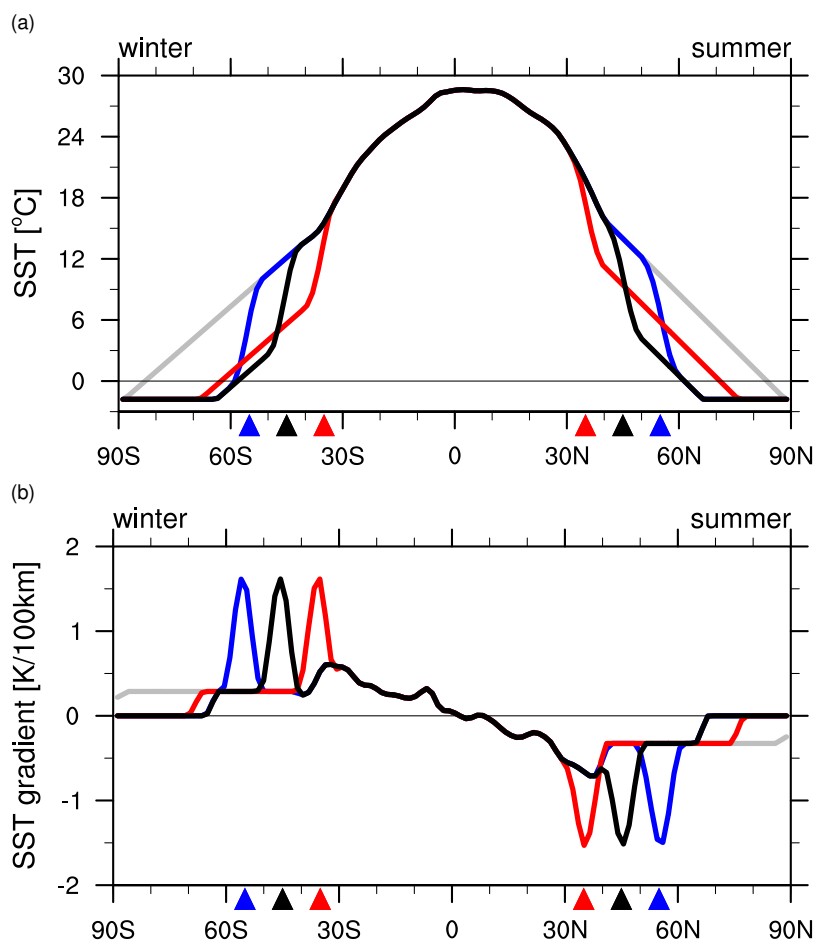

**Figure 1.** (a) Meridional profiles of the prescribed zonally symmetric sea surface temperature [°C] and (b) their meridional gradient [K $10^{-2}$ km$^{-1}$]. Colors (red, black, blue) indicate the latitude of the SST front (35°,45°,55°) as indicated by the triangles. The grey line corresponds to the non-front reference (REF) experiment.



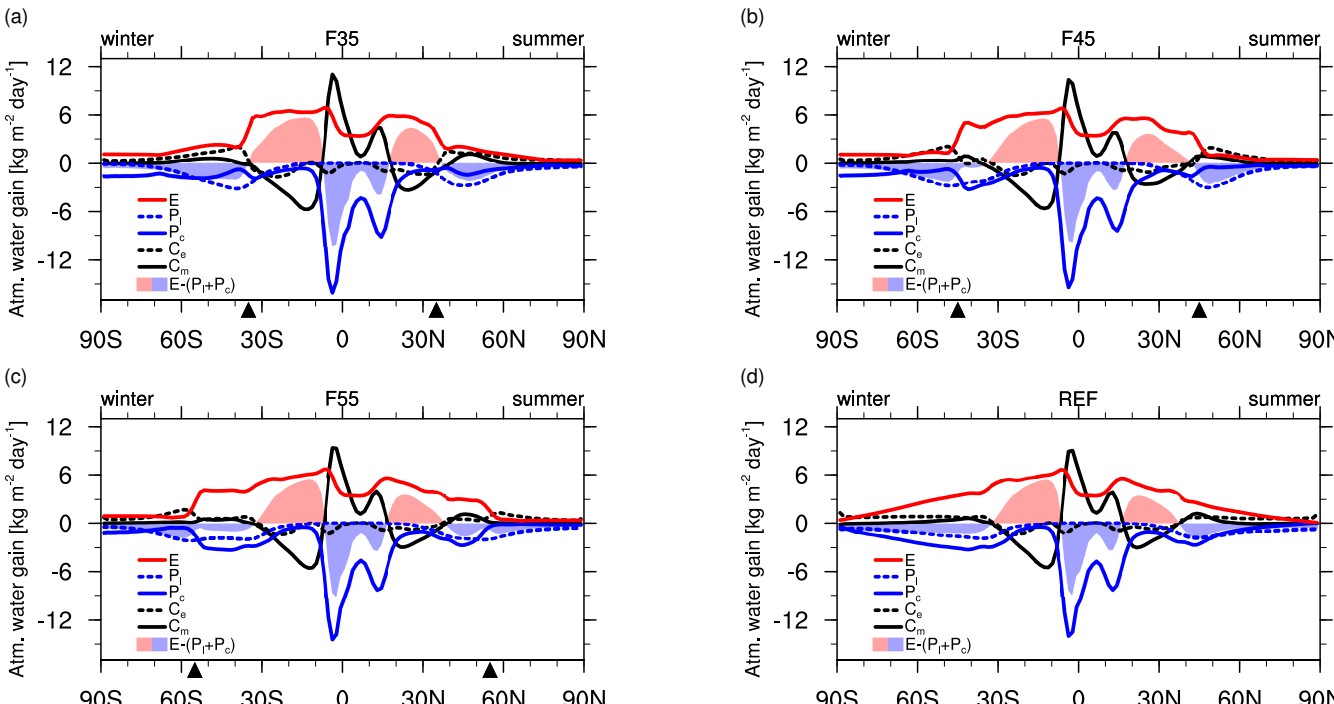

**Figure 2.** Meridional diagram of the simulated zonal mean evaporation (red line), large-scale precipitation (blue dashed line), cumulus precipitation (blue solid line), vertically integrated convergence of northward eddy moisture flux (black dashed line), vertically integrated convergence of northward mean-flow moisture flux (black solid line), evaporation minus precipitation (shading, red color is for positive values and blue color is for negative values, respectively) for (a) F35, (b) F45, (c) F55 and (d) REF experiments. Unit is [kg m$^{-2}$ day$^{-1}$].



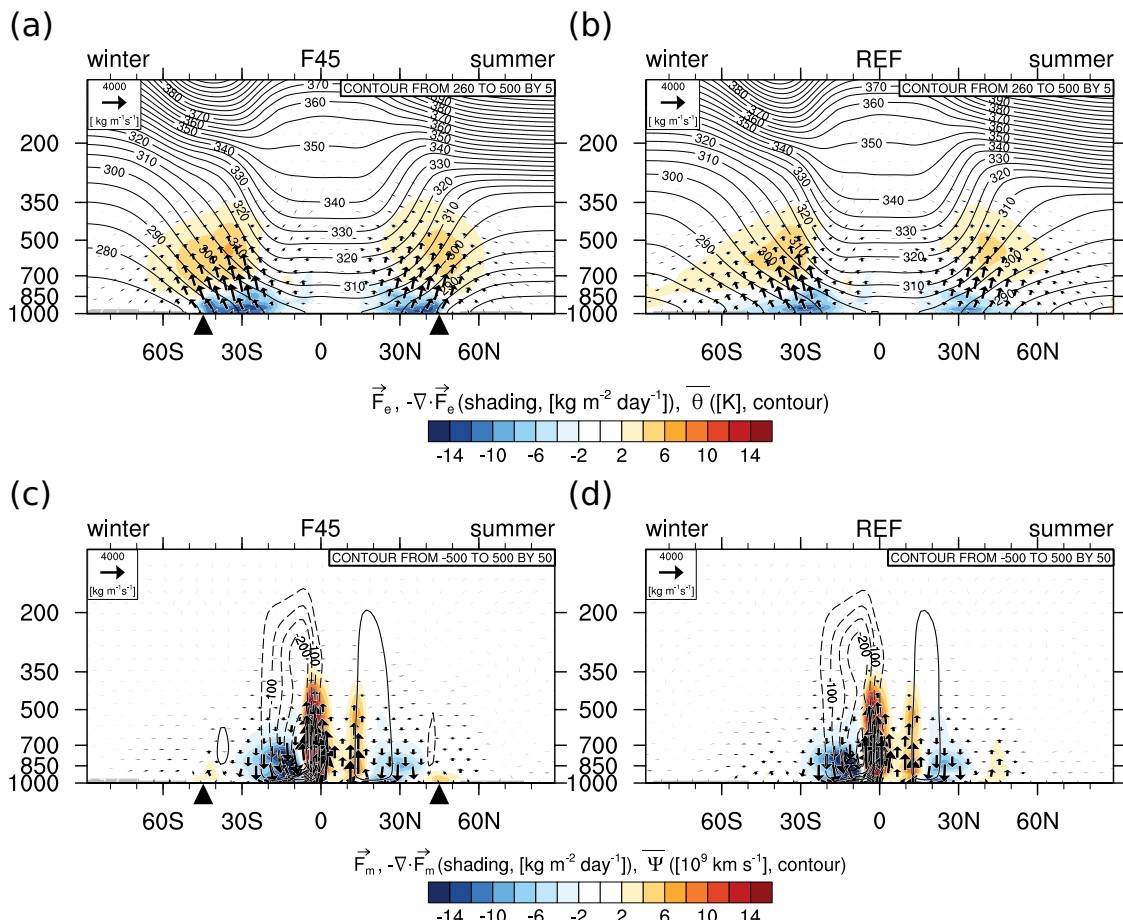

**Figure 3.** (a) Simulated zonally averaged mean eddy moisture flux (vector, [kg m$^{-2}$ day$^{-1}$]), its convergence (shading, [kg m$^{-3}$ day$^{-1}$]), and potential temperature (contour, [K]) for F45. (c) Zonally averaged mean-flow moisture flux (vector), its convergence (shading), and mass-stream function (contour, [10$^9$ kg s$^{-1}$]) for F45. (b,d) Same as (a,c), but for REF.






**Figure 4.** (a) Mean moistening tendency due to micro-physics parameterization (shading, [kg m$^{-3}$ day$^{-1}$]), potential temperature (black contour, [K]), and zonal wind (brown contours for a contour interval of 15 [m s$^{-1}$] from 15 [m s$^{-1}$]) for F45. (c) Mean moistening tendency due to cumulus parameterization (shading) and specific humidity (contour, [g kg$^{-1}$]) for F45. (e) Mean moistening tendency due to diffusive process (shading) and specific humidity (contour) for F45. (b,d,f) Same as (a,c,e), but for REF.



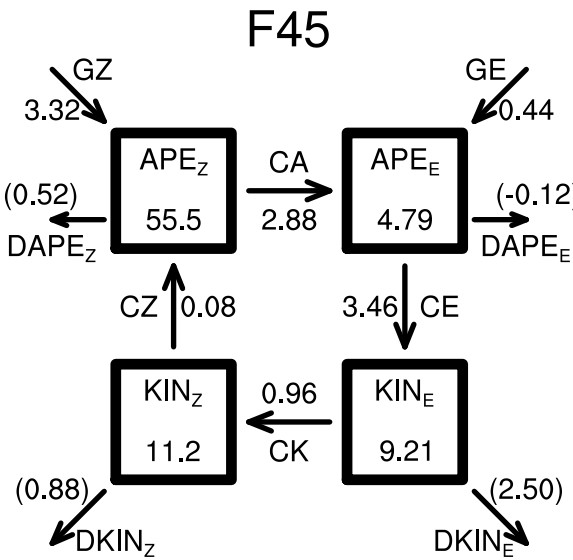

**Figure 5.** A schematic diagram illustrating the Lorenz energy cycle for F45 as calculated as the zonally and vertically-weighted average from 850hPa to 200hPa. Arrows indicate the direction of the positive energy conversion or generation. APE (KIN) stands for the available potential energy (kinetic energy), respectively. Subscripts Z and E are for the zonal mean state and eddies, respectively. Values with parentheses are calculated as the residual to close the budget. The Unit for APE and KIN is [$10^5$ J m$^{-2}$], and [$10^5$ J m$^{-2}$day$^{-1}$] for the rest of the calculation. See text for more details.





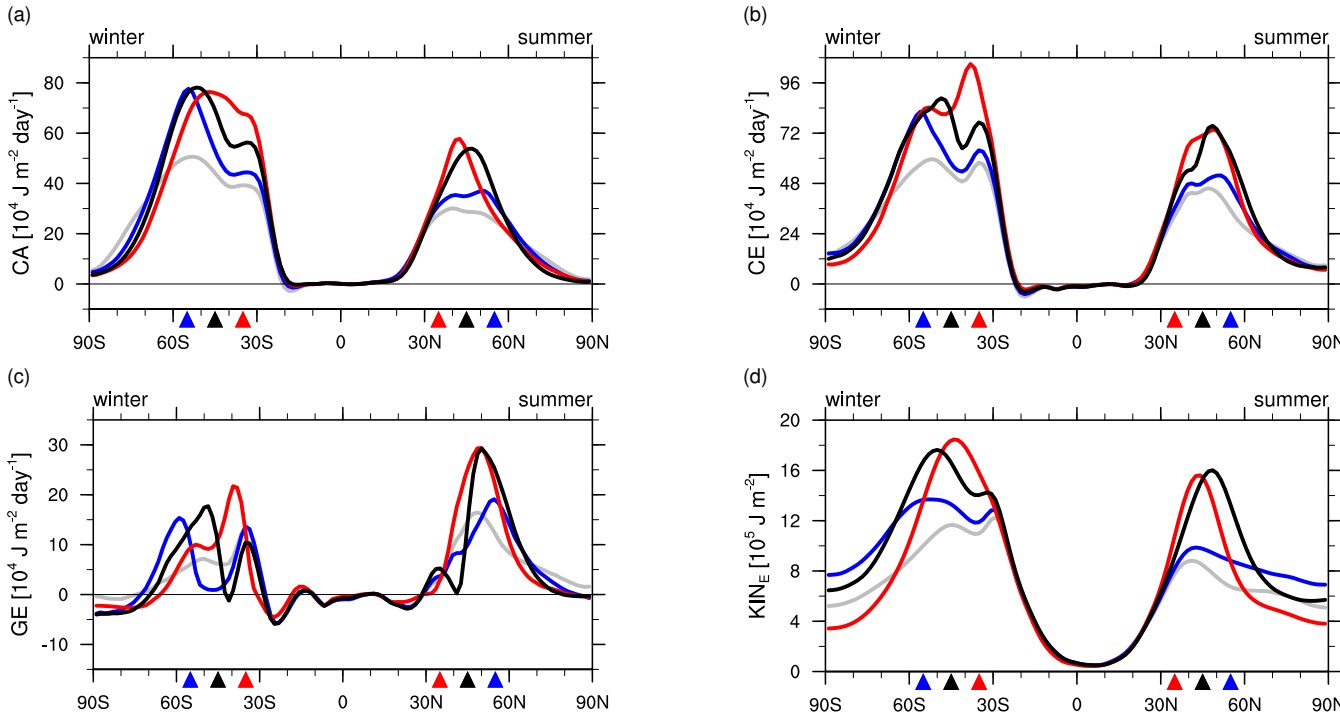

**Figure 6.** (a) CA, (b) CE, (c) GE, and (d) $KIN_E$ averaged zonally and vertically from 850hPa to 200hPa for F35(red), F45(black), F55(blue), REF(gray), respectively.



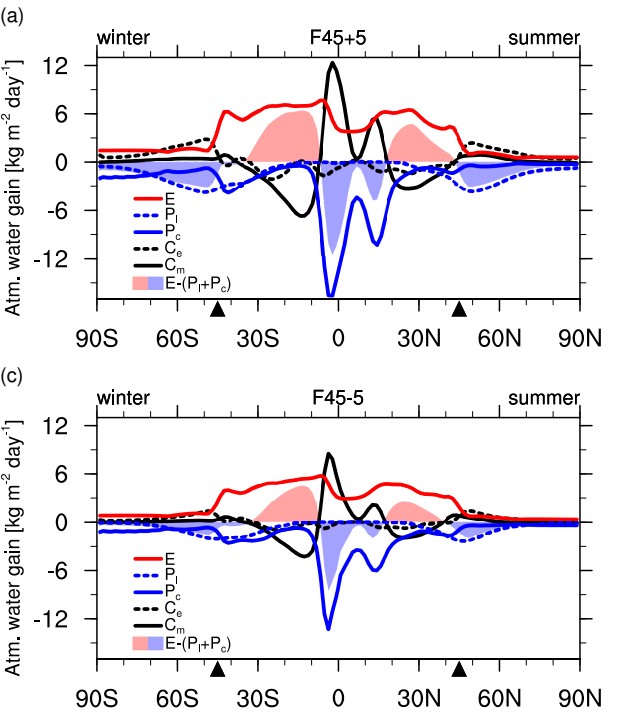

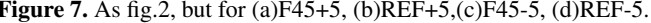

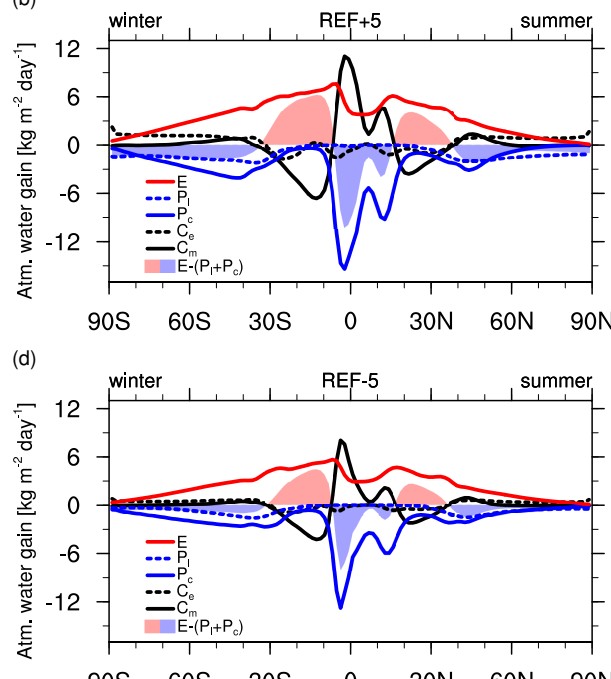

**Figure 7.** As fig.2, but for (a)F45+5, (b)REF+5,(c)F45-5, (d)REF-5.





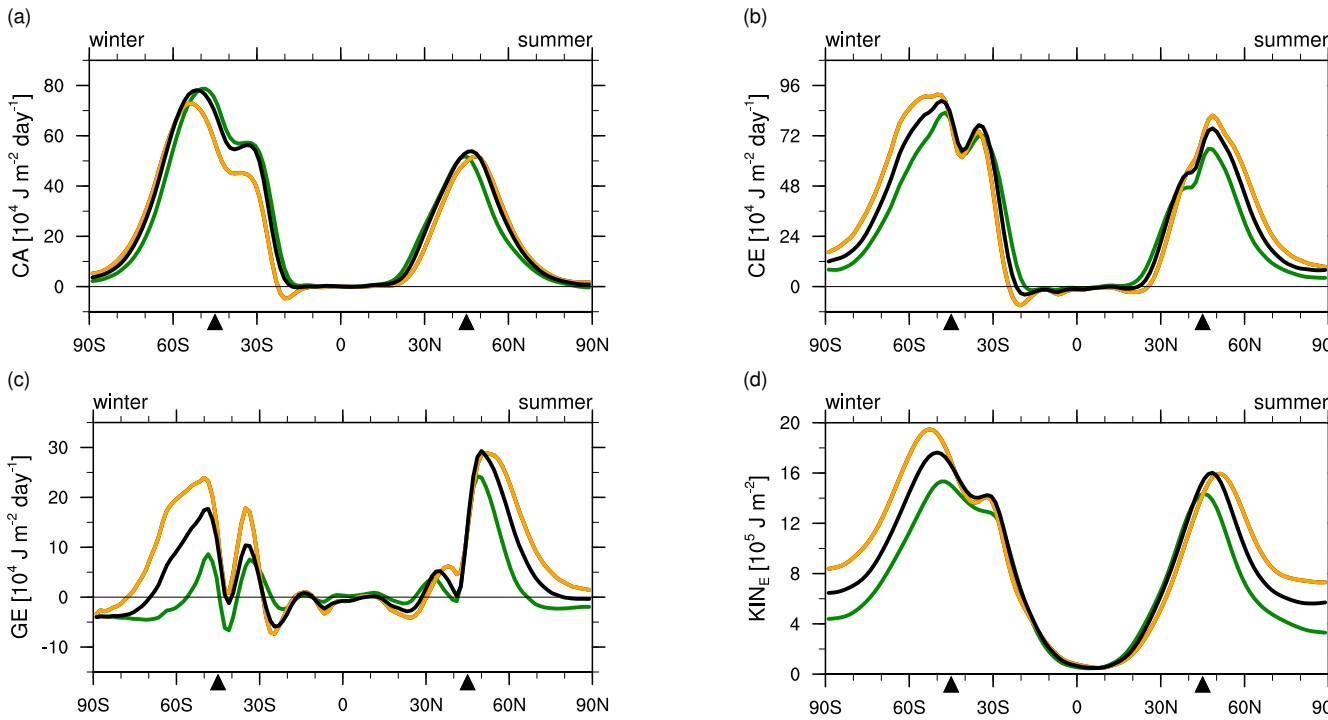

**Figure 8.** as fig. 6, but for F+45 (orange), F45 (black) and F-45 (green).