# Peer review of "Influence of mid-latitude Sea Surface Temperature Fronts on the Atmospheric Water Cycle and Storm Track Activity"

_EGUsphere, 2024_

## Referee Comment (RC1)

Review on the WCD manuscript egusphere-2024-735, "Influence of mid-latitude Sea Surface Temperature Fronts on the Atmospheric Water Cycle and Storm Track Activity"

General comment:

In this paper, model output of "aqua-planet" experiments was examined to illustrate the effects of a midlatitude SST front on atmospheric water cycle and storm track activity. The position of a midlatitude SST front was found to have noticeable influences on surface evaporation, atmospheric moisture fluxes and precipitation. Storm track activity was changed as well, by the change of the SST frontal latitude, through the changes in eddy energy conversions and generation of eddy available potential energy by diabatic heating.

I think that the analyses of atmospheric water cycle and energy cycle of transient eddies in this paper were done systematically, and that the presented results and conclusion are reasonable in general. However, some terms were used in the analysis without sufficient descriptions, which sometimes hinders reader's understanding of the details of the results. I therefore recommend minor revision of this manuscript.

Specific comments:

1. Section 4, L119-L133
(a) CZ is missing in the equations. (Presumably, Eq. 14)
(b) The meaning of CE, CZ, CA, CK, GE, and GZ should be described. (Conversion from what/to what, and generation of what kind of energy?)

2. In section 7, the roles of $\dot{q}_m$, $\dot{q}_c$, $\dot{q}_b$ in the model should be described to clarify the physical processes those terms represent. Does $\dot{q}_m$ only represent the removal of moisture by large-scale condensation? Does $\dot{q}_c$ correspond to both of the removal of moisture by convective precipitation and the vertical redistribution of moisture by convection? Does $\dot{q}_b$ correspond only to the vertical redistribution of moisture evaporated from the ocean surface? Without the understanding of the physical processes that those tendency terms represent, it is difficult to interpret Fig. 4.

3. (L243-L244) "the double peak structure in GE is hinted in both CA and KIN$_E$,"
Why is the double peak structure reflected in CA? In Figure 5, GE adds energy to the eddy APE, but CA is at the *upstream* side of the energy flow to the eddy APE (CA supplies energy to the eddy APE) in the energy cycle.

---

## Author Comment (AC1)

Summary of the reviewers' comments are written using black font. Responses to our reviewers are written using in blue font.

Reviewer #1 (RC1)

Review on the WCD manuscript egusphere-2024-735, "Influence of mid-latitude Sea Surface Temperature Fronts on the Atmospheric Water Cycle and Storm Track Activity" General comment: In this paper, model output of "aqua-planet" experiments was examined to illustrate the effects of a midlatitude SST front on atmospheric water cycle and storm track activity. The position of a midlatitude SST front was found to have noticeable influences on surface evaporation, atmospheric moisture fluxes and precipitation. Storm track activity was changed as well, by the change of the SST frontal latitude, through the changes in eddy energy conversions and generation of eddy available potential energy by diabatic heating. I think that the analyses of atmospheric water cycle and energy cycle of transient eddies in this paper were done systematically, and that the presented results and conclusion are reasonable in general. However, some terms were used in the analysis without sufficient descriptions, which sometimes hinders reader's understanding of the details of the results. I therefore recommend minor revision of this manuscript.

Thank you very much for dedicating your time to reviewing our manuscript. Your valuable and constructive comments help to improve the quality of our work.

Specific comments:
1. Section 4, L119-L133
(a) CZ is missing in the equations. (Presumably, Eq. 14)

We have added the equations.

(b) The meaning of CE, CZ, CA, CK, GE, and GZ should be described. (Conversion from what/to what, and generation of what kind of energy?)

We have added the descriptions.

2. In section 7, the roles of $q'_m$, $q'_c$, and $q'_b$ in the model should be described to clarify the physical processes those terms represent. Does $q'_m$ only represent the removal of moisture by large-scale condensation? Does $q'_c$ correspond to both of the removal of moisture by convective precipitation and the vertical redistribution of moisture by convection? Does $q'_b$ correspond only to the vertical redistribution of moisture evaporated from the ocean surface? Without the understanding of the

physical processes that those tendency terms represent, it is difficult to interpret Fig. 4.

We have clarified the descriptions of $\dot{q}_m$, $\dot{q}_c$, and $\dot{q}_b$.

3. (L243-L244) "the double peak structure in GE is hinted in both CA and $KIN_E$," Why is the double peak structure reflected in CA? In Figure 5, GE adds energy to the eddy APE, but CA is at the upstream side of the energy flow to the eddy APE (CA supplies energy to the eddy APE) in the energy cycle.

As the reviewer pointed out, CA is indeed upstream of $APE_E$. However, we would like to draw the reviewer's attention to the fact that CA and GE are pointing in parallel toward $APE_E$, where CA is not independent of GE, because T' in CA is partly due to Q' in GE. In other words, CA includes an indirect effect of diabatic heating and, therefore, may be amplified through GE. We modified the sentence for clarification.

Reviewer #2 (CC1)

Review of Ogawa and Spengler, "Influence of mid-latitude Sea Surface Temperature fronts on the atmospheric water cycle and storm track activity" submitted to EGUsphere.

This paper investigates the effect of diabatic processes at SST fronts on the water cycle and Extratropical storm track. The tool is an aqua-planet model which has been often used in previous studies, but still provides some useful insight. By necessity, the experiments are idealized. The analysis, of moisture budgets and Lorenz energy cycle, is sophisticated and thorough, and the interpretation very reasonable. They find that the latitude of the SST front plays a key role in determining the Extratropical water cycle and storm track, extending previous work by Sampe, H. Nakamura and Ogawa that focused on more basic storm track metrics. The hydrologic cycle is a key aspect of climate state and climate variability, and I think this novel work should be published subject to the revisions suggested below. I indicate Major Revisions because it may take some time to address some of my points including figure re-plotting, but I do not have major disagreements on the main points.

Justin Small, NCAR

Thank you very much for dedicating your time to reviewing our manuscript. Your valuable and constructive comments helped to greatly improve the quality of our work.

Main Comments

Line 45 and line 225-227. The low level baroclinicity is not really discussed further in the paper. Can you say something about the relative role of the diabatic processes discussed in this paper vs the baroclinicity? Can you relate it to the Lorenz energy cycle, perhaps?

Thank you for pointing this out. We have included a brief discussion of the increase in low-level baroclinicity in response to the existence of the SST front in section 6. Regarding the Lorenz energy calculation. It requires that potential temperature surfaces are not overturning vertically. In other words, the isentropes need to monotonously increase in the vertical, implying a positive static stability. This requirement is not fulfilled in the boundary layer, and we are hence precautionarily excluding levels below 850 hPa from these calculations and our analysis.

Line 251. I am thinking that your global "+/-5K" simulations also significantly warm/cool the Tropics,

which may teleconnect to Extratropics. Can you think of an experiment to focus on SST changes in Extratropics only? (This may be difficult as it is likely to add an artificial meridional gradient of SST).

We agree with the reviewer's concerns and have considered alternative experimental designs. However, as the reviewer indicated, it is very difficult to find a suitable experimental design and we hence used these previously used and published profiles. We are currently working on revised zonal mean profiles to further address this issue in future publications.

An experimental setting with uniform SST changes only at mid- and high-latitudes without changing tropical water temperatures is unattainable without changing the low-latitude SST gradient. Such changes in the low-latitude SST gradient could considerably change the climatological mean latitude and intensity of the thermally driven subtropical jet, which would change the waveguide and baroclinicity in the subtropical troposphere (e.g. Nakamura and Sampe 2002) as well as the stationary response of the atmospheric circulation, thus complicating the issues addressed in this study of mid-latitude disturbances. We chose the experimental setup of +/-5K to avoid these secondary effects.

Other comments

Fig. 2. The line plots are all dominated by the signal in the Tropics, and this is quite similar between cases, as you describe. You could have one panel with the full fields (F45 or REF), and other panels are differences from that case.

The diagrams suggested by the reviewer are shown as Figure R1. Here, only the result from REF is shown in full fields and the results of the other experiments are shown as differences from REF. Even though the changes in the hydrological cycle mostly show the shift associated with the ocean frontal as well a slight widening of the Hadley cell, the direct link of the features in the climatological hydrological cycle are more difficult to directly associate with the SST front. Hence, we propose to keep the full fields for easier interpretation.

Line 2. "implies a crucial role"-> "suggests". The first sentence only noted a spatial correspondence which is not conclusive in itself.

We fixed the expression from "implies" to "suggests." In addition, we joined the first two sentences to become more conclusive.

Line 18 "maximize in the Extratropics" – as you must be excluding the ITCZ in this sentence.

We fixed it accordingly.

Line 58. Add a sentence describing convective and large-scale parameterizations, as these are analyzed in the paper.

We added the descriptions and corresponding citations.

Equations 9-22 are quite extensive! In my copy, I did not see where CZ is defined. Maybe Q is not defined, and DAPE and DKIN.

There are certainly a lot of formulas, but we think they should be listed. We added an equation for CZ (14) and clarified the definition of Q. The meaning of DAPE and DKIN, which are calculated as the residual to close the budget, are described after equation (22).

Line 138. In practice, there is usually some small trend, either drift or interannual variability.

We changed the expression from "must vanish" to "is approximately zero".

Lines 175-179. Please specify where you are talking about Fig. 3a or Fig. #c.

We added the figure reference.

Line 181 For reference to $C_m$, should this be" (black line, Fig. 2b)"

We added the figure reference.

Line 190. "(Fig. 2b)" ?

We fixed it.

Lines 191-197. Consider first discussing the REF experiment, which has biggest difference to F45, before discussing the other latitude cases. You might also point out that F_m looks very similar between F45 and REF.

We have considered changing the order of the discussion as proposed by the reviewer. As this paragraph is located right after the paragraph discussing the results from F45, we think it is more straightforward to discuss the experiments with the presence of an SST front at different latitudes first, followed by the discussion of the reference experiment (REF). Therefore, we would like to keep the order of the discussion. We agree with the reviewer's point about the similarity of F_m between F45 and REF. We added a sentence to explicitly point this out.

Line 199. Convergence of F -> "(Fig. 3a and Fig. 3c)" i.e. F is the total.

We added the reference to Fig. 3c.

Fig. 8. Typos in caption, should be F45+5 etc.

We fixed it.

The shaded fields in Fig. 3 and 4 are all parts of the moisture budget. Would it be useful to show all on the same color bar, as they should balance? This may be difficult as I note Figs 4e,f are on a much larger range color bar (meaning there must be substantial cancellation, probably by convective param., which presumably saturates the color bar).

We understand the reviewer's point. Actually, we already tested exactly what the reviewer proposed before submitting the previous version. As a result, we faced the saturation of the color bar, in alignment with the reviewer's concern. Therefore, we decided to use several different color bars and would like to keep the color bars of these figures.

Line 229. 850hPa to 200hPa – why not from surface?

The Lorenz energy calculation requires that potential temperature surfaces are not overturning vertically. In other words, the isentropes need to monotonously increase in the vertical, implying a positive static stability. This requirement is not fulfilled in the boundary layer and we are hence precautionarily excluding levels below 850 hPa from these calculations and our analysis.

Line 238. CA does change significantly in the REF experiment – I think this is consistent with Sampe et al. 2010, and should be mentioned.

We agree with the reviewer and added a sentence accordingly.

Line 246. "in winter but not in summer (Fig. 6c)"

We agree with the reviewer's comment and added this phrase accordingly.

Fig. 7 – consider showing as differences, F45+5 minus F45 etc.

We would like to keep Fig. 7 unchanged, see our response to the first "other comments" by the reviewer, because we would like to keep the same format to be comparable with Fig. 2.

Fig. 8a – CA does not change much because the baroclinicity is not changed when adding/subtracting a fixed SST?

We agree with the reviewer and added this point.